# Trends in suicide deaths before and after the COVID-19 outbreak in Korea

**Seunghyong Ryu**[1], **Hee Jung Nam**[2], **Min Jhon**[1], **Ju-Yeon Lee**[1,3], **Jae-Min Kim**[1], **Sung-Wan Kim**[1,3]*

**1** Department of Psychiatry, Chonnam National University Medical School, Gwangju, Korea, **2** Department of Psychiatry, Seoul Medical Center, Seoul, Korea, **3** Mindlink, Gwangju Bukgu Mental Health Center, Gwangju, Korea

* swkim@chonnam.ac.kr

## Abstract

We investigated the effect of the coronavirus disease-2019 (COVID-19) pandemic on suicide trends in Korea via a time-series analysis. We used Facebook Prophet to generate forecasting models based on the monthly numbers of suicide deaths in Korea between 1997 and 2018, validated the models by comparison with the 2019 numbers, and predicted the numbers of suicides in 2020. We compared the expected and observed numbers of suicides during the COVID-19 pandemic. The total numbers of suicides during the COVID-19 pandemic did not deviate from projections based on the pre-pandemic period. However, the number of suicides among women and those under the age of 34 years significantly exceeded the expected level. The COVID-19 pandemic did not increase the overall suicide rate significantly. However, suicides among women and young people increased, suggesting that the pandemic might drive more members of these groups to suicide. Further studies are needed to verify the long-term impact of the COVID-19 pandemic on suicide.

## 1. Introduction

Since early 2020, the coronavirus disease 2019 (COVID-19) pandemic has been a public health crisis and has overwhelmed people's daily lives globally [1]. People are fearful of the high infectivity and mortality of COVID-19 and restrict their daily activities, such as meeting family and friends, exercising, and visiting hospitals [2]. To curtail the spread of COVID-19, authorities have implemented stringent social distancing policies, including restricting public gatherings, closing schools, and requiring the wearing of face masks in public facilities [3]. The prolonged pandemic has aggravated financial difficulties for low-income earners due to increased unemployment and reduced income [4]. As a result, people continue to experience stress while adapting to the extreme circumstances of the pandemic [5, 6].

Given this situation, experts are concerned about the deterioration of people's mental health and the consequent increase in suicide rate [7]. Psychological distress due to isolation, loneliness, bereavement, increased alcohol consumption, financial stressors, and disrupted health care are major factors that could increase the suicide risk during the COVID-19 pandemic [8]. However, recent research could not find evidence of an increase in suicide rates

**Data Availability Statement:** All data in this paper are publicly available at https://mdis.kostat.go.kr, Microdata Integrated Service provided by the Statistics Korea. Please refer to https://mdis.kostat.

go.kr/eng/pageLink.do?link=mdisService for information on how to access it.

**Funding:** This research was supported by grants of the Korea Health Technology R&D Project through the Korea Health Industry Development Institute (KHIDI), funded by the Ministry of Health & Welfare, Republic of Korea (grants number: HI19C0481, HC19C0316). The funders were not involved in the conception, design, analysis or interpretation of this study.

**Competing interests:** The authors have declared that no competing interests exist.

during the pandemic [9, 10]. A recent study of global changes in suicide trends before and after the COVID-19 outbreak showed that suicide numbers remained largely unchanged or even declined in high- and upper-middle-income countries [11]. This interrupted time-series analysis found statistical evidence of a declining trend in suicides in the early months of the pandemic in Korea. However, this finding was calculated from preliminary data for a short period in the early months of the pandemic, and care should be taken in its interpretation.

Although the prevalence of COVID-19 in Korea has been lower than other countries and the Korean government did not institute a complete lockdown, the social distancing policies and restricted economic activities have persisted [12]. Furthermore, the stress, anxiety, and depression levels of Koreans during the pandemic were greatly increased compared with before the pandemic, as in other countries [13, 14]. Before the COVID-19 outbreak, suicide was one of the most serious public health and social issues in Korea, where the suicide rate is the highest among Organization for Economic Cooperation and Development countries [15, 16]. Therefore, suicide trends during the COVID-19 pandemic in Korea need to be closely monitored. Accordingly, this study investigated changes in suicide incidence before and after the COVID-19 outbreak in Korea in more detail, examining the difference between the number of suicides during the pandemic (February to December 2020) and the projections based on data obtained over the past two decades.

## 2. Materials and methods

### 2.1. Data sources and study population

Using the cause of death statistics provided by Statistics Korea's Microdata Integrated Service, we obtained data on the sex, age, and date of death of 283,633 suicide victims between 1997 and 2020 [17]. The codes assigned to the cause of death due to suicide were X60–X84 according to the International Statistical Classification of Diseases and Related Health Problems 10th Revision. From these data, we calculated the number of suicides per month by sex and age group ($\leq 34$, 35–49, 50–64, and $\geq 65$ years).

This study was approved by the Institutional Review Board of Chonnam National University Hospital (IRB number: CNUH-EXP-2021-047).

### 2.2. Time series analysis

From data on suicide deaths each month between 1997 and 2019, 264 observations up to December 2018 were used to fit the time series models forecasting the next 24 months. The 12 observations for 2019 were used to validate the models.

We used Facebook Prophet, a time series forecasting procedure based on an additive model in which non-linear trends are fitted with seasonality and holiday effects [18]. Prophet is as sophisticated as the autoregressive integrated moving average (ARIMA) model, a widely used forecasting procedure [19]. Prophet has the advantage of being able to adjust for the effects of holidays and other recurring events, which is why we used the Prophet algorithm to adjust for the effect of celebrity suicides around the COVID-19 outbreak. First, using the Prophet algorithm, we fitted a time series model based on data for 264 months between 1997 and 2018 with multiplicative seasonality mode. The effect of 19 celebrity suicides was modelled with extra regressors. Then, we tuned the hyperparameter to match the observed and expected numbers in 2019 as closely as possible in terms of the root mean squared error, mean absolute error, and mean absolute percentage error, and made predictions with a 24-month forecasting period (the expected number of suicides per month between 2019 and 2020). Additionally, when comparing accuracy for predicting suicides in 2019 between the Prophet and ARIMA algorithms, we determined that the fitted Prophet models performed comparably to, or better

than, the ARIMA models (S1 Table). Finally, we visualized the observed and expected number of suicides.

We compared this expected number to the observed number of suicides during the COVID-19 pandemic (February to December 2020) by calculating observed-to-expected ratios (OERs) and confidence intervals (CIs). Statistical significance was defined as a 95% CI that excluded the null value of 1.00. All analyses were conducted using R (ver. 4.0.3; R Development Core Team, Vienna, Austria).

## 3. Results

### 3.1. Observed and expected numbers of suicides per month during the COVID-19 pandemic

Figs 1–3 depict the number of suicide deaths per month between 2017 and 2020. Tables 1–3 show the observed and expected numbers of suicides per month during the pandemic, together with OERs and 95% CIs. The total number of suicides per month during the COVID-19 pandemic did not deviate overall from the projected estimates (Fig 1 and Table 1). The monthly number of male suicides was slightly lower than expected (Fig 2(A)), but the difference was not significant (Table 2). However, the monthly number of female suicides greatly exceeded the expected level (Fig 2(B) and Table 2). Compared to the projections, the actual

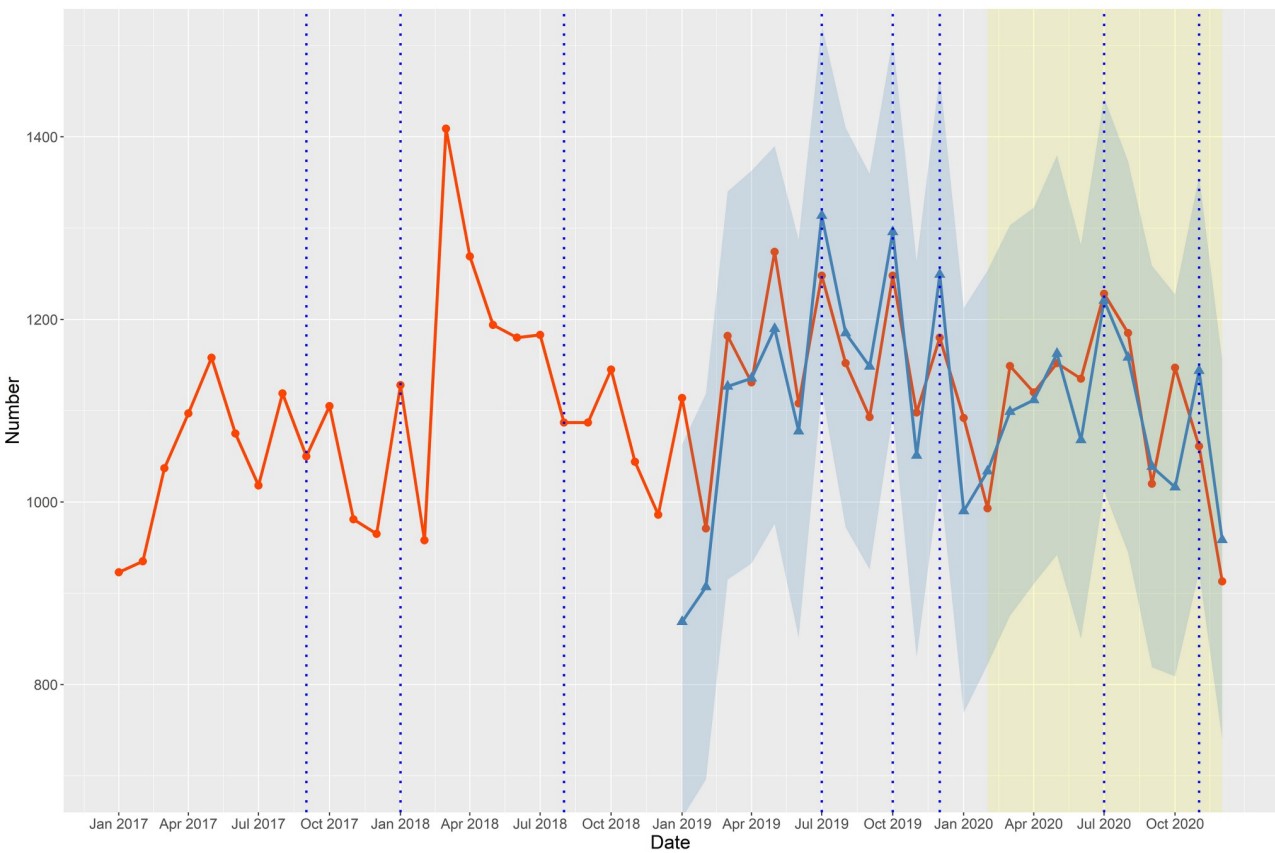

**Fig 1. Observed and expected numbers of suicides per month between 2017 and 2020 in Korea.** The red line with circles represents the observed number of suicides between 2017 and 2020. The blue line with triangles represents the expected number of suicides according to forecasting models and the surrounding blue area indicates the 95% confidence interval (with 2019 as the validation period). The blue vertical dotted lines indicate the dates of celebrity suicides.

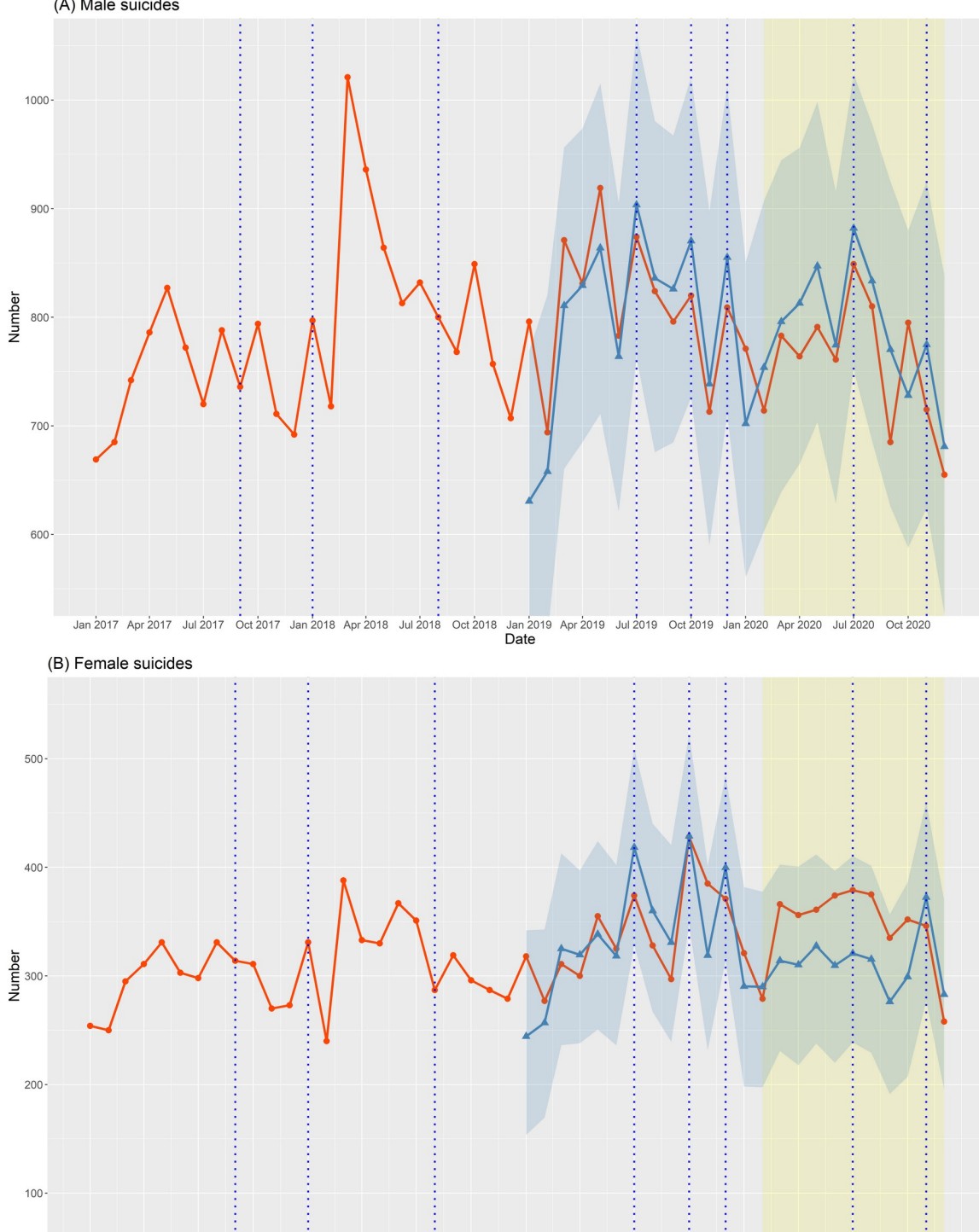

**Fig 2. Observed and expected numbers of male and female suicides per month between 2017 and 2020 in Korea.** The red line with circles represents the observed number of suicides between 2017 and 2020. The blue line with triangles represents the expected number of suicides according to forecasting models and the surrounding blue area indicates the 95% confidence interval (with 2019 as the validation period). The blue vertical dotted lines indicate the dates of celebrity suicides.

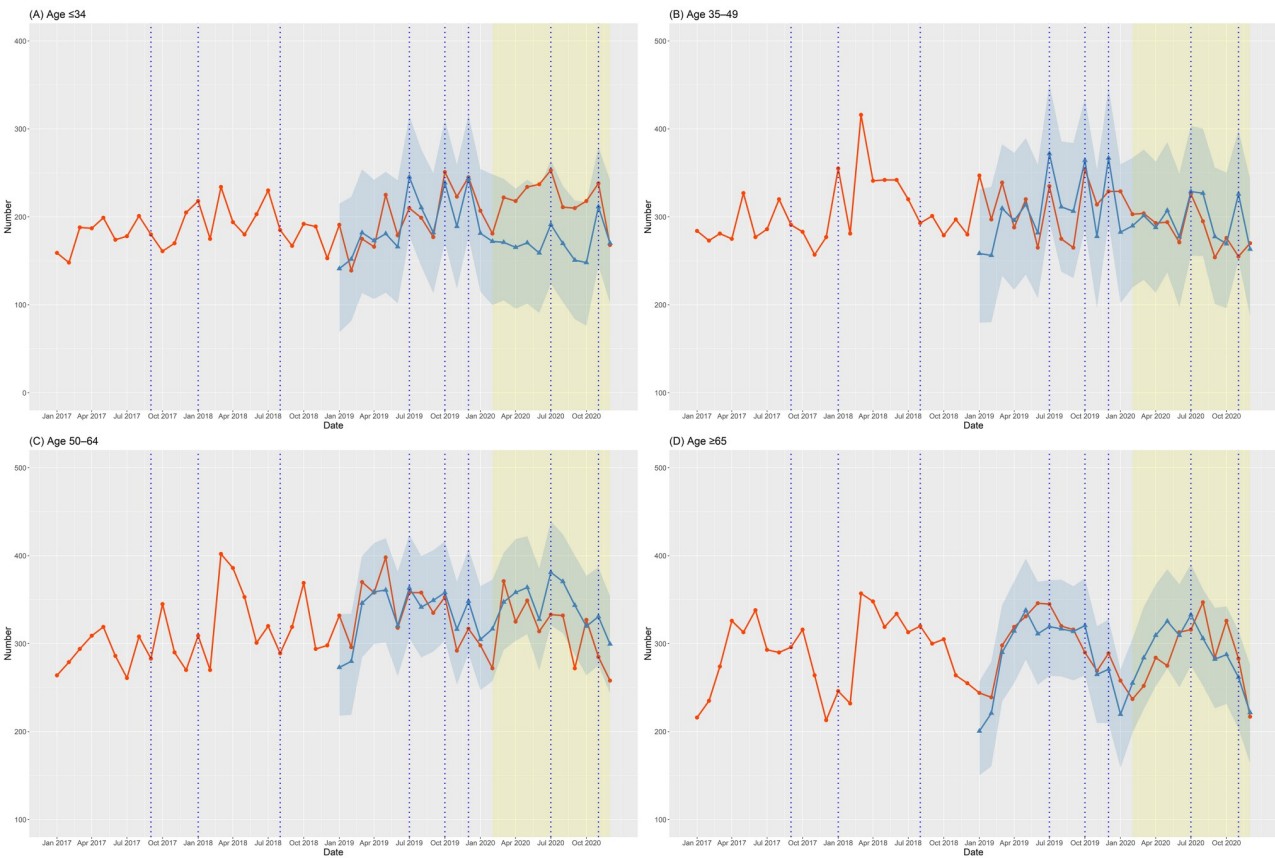

**Fig 3. Observed and expected numbers of suicides by age group per month between 2017 and 2020 in Korea.** The red line with circles represents the observed number of suicides between 2017 and 2020. The blue line with triangles represents the expected number of suicides according to forecasting models and the surrounding blue area indicates the 95% confidence interval (with 2019 as the validation period). The blue vertical dotted lines indicate the dates of celebrity suicides.

**Table 1. Number of suicides per month during the COVID-19 pandemic in Korea.**

|  | Observed | Expected (95% CI) | OER (95% CI) |
|---|---|---|---|
| Feb | 993 | 1,033.67 (822.03–1,253.58) | 0.96 (0.90–1.02) |
| Mar | 1,149 | 1,098.95 (875.38–1,303.30) | 1.05 (0.99–1.11) |
| Apr | 1,120 | 1,111.68 (910.28–1,322.49) | 1.01 (0.95–1.07) |
| May | 1,152 | 1,162.50 (942.11–1,379.77) | 0.99 (0.93–1.05) |
| Jun | 1,135 | 1,067.91 (850.25–1,281.66) | **1.06 (1.00–1.12)** |
| Jul | 1,228 | 1,220.71 (1012.15–1,443.11) | 1.01 (0.95–1.06) |
| Aug | 1,185 | 1,158.28 (944.42–1,373.32) | 1.02 (0.96–1.08) |
| Sep | 1,020 | 1,038.64 (818.93–1,258.88) | 0.98 (0.92–1.04) |
| Oct | 1,147 | 1,016.27 (808.98–1,227.47) | **1.13 (1.06–1.19)** |
| Nov | 1,061 | 1,143.76 (922.67–1,357.29) | 0.93 (0.87–0.98) |
| Dec | 913 | 958.46 (740.14–1,155.45) | 0.95 (0.89–1.01) |

Bold indicates a statistically significant OER with a 95% CI that excludes the null value of 1.00.

Abbreviations: OER, Observed-to-Expected Ratio; CI, Confidence Interval.

**Table 2. Number of male and female suicides per month during the COVID-19 pandemic in Korea.**

| | Men | | | Women | | |
|---|---|---|---|---|---|---|
| | Observed | Expected (95% CI) | OER (95% CI) | Observed | Expected (95% CI) | OER (95% CI) |
| Feb | 714 | 753.84 (602.94–907.40) | 0.95 (0.88–1.02) | 279 | 289.99 (197.48–377.51) | 0.96 (0.85–1.07) |
| Mar | 783 | 796.02 (639.15–944.73) | 0.98 (0.91–1.05) | 366 | 314.10 (230.87–402.58) | **1.17 (1.05–1.28)** |
| Apr | 764 | 812.96 (664.94–956.05) | 0.94 (0.87–1.01) | 356 | 310.28 (217.86–400.68) | **1.15 (1.03–1.27)** |
| May | 791 | 847.27 (703.44–998.12) | 0.93 (0.87–1.00) | 361 | 327.66 (237.80–411.82) | 1.10 (0.99–1.22) |
| Jun | 761 | 774.45 (628.74–915.65) | 0.98 (0.91–1.05) | 374 | 309.65 (220.11–396.87) | **1.21 (1.09–1.33)** |
| Jul | 849 | 881.98 (752.26–1,024.15) | 0.96 (0.90–1.03) | 379 | 320.80 (238.81–410.32) | **1.18 (1.06–1.30)** |
| Aug | 810 | 833.53 (686.18–977.81) | 0.97 (0.90–1.04) | 375 | 315.40 (229.32–401.42) | **1.19 (1.07–1.31)** |
| Sep | 685 | 770.31 (625.85–925.09) | 0.89 (0.82–0.96) | 335 | 276.24 (191.23–356.64) | **1.21 (1.08–1.34)** |
| Oct | 795 | 728.10 (587.92–880.08) | **1.09 (1.02–1.17)** | 352 | 299.22 (207.24–386.25) | **1.18 (1.05–1.30)** |
| Nov | 715 | 774.86 (624.23–924.99) | 0.92 (0.86–0.99) | 346 | 372.46 (277.15–460.37) | 0.93 (0.83–1.03) |
| Dec | 655 | 680.92 (527.01–838.26) | 0.96 (0.89–1.04) | 258 | 282.94 (195.56–369.58) | 0.91 (0.80–1.02) |

Bold indicates a statistically significant OER with a 95% CI that excludes the null value of 1.00.

Abbreviations: OER, Observed-to-Expected Ratio; CI, Confidence Interval.

**Table 3. Number of suicides by age group per month during the COVID-19 pandemic in Korea.**

| | Age ≤ 34 | | | Age 35–49 | | | Age 50–64 | | | Age ≥ 65 | | |
|---|---|---|---|---|---|---|---|---|---|---|---|---|
| | Observed | Expected (95% CI) | OER (95% CI) | Observed | Expected (95% CI) | OER (95% CI) | Observed | Expected (95% CI) | OER (95% CI) | Observed | Expected (95% CI) | OER (95% CI) |
| Feb | 181 | 172.14 (99.73–248.33) | 1.05 (0.90–1.20) | 303 | 289.80 (219.87–366.97) | 1.05 (0.93–1.16) | 272 | 316.80 (257.66–372.72) | 0.86 (0.76–0.96) | 237 | 255.23 (198.77–305.56) | 0.93 (0.81–1.05) |
| Mar | 222 | 171.11 (105.09–243.24) | **1.30 (1.13–1.47)** | 304 | 301.52 (228.42–376.54) | 1.01 (0.89–1.12) | 371 | 347.30 (293.01–403.07) | 1.07 (0.96–1.18) | 252 | 283.93 (225.55–341.57) | 0.89 (0.78–1.00) |
| Apr | 218 | 165.34 (95.65–232.11) | **1.32 (1.14–1.49)** | 293 | 287.87 (213.93–362.82) | 1.02 (0.90–1.13) | 325 | 358.22 (303.14–418.89) | 0.91 (0.81–1.01) | 284 | 309.47 (251.48–367.19) | 0.92 (0.81–1.02) |
| May | 234 | 170.51 (101.35–242.26) | **1.37 (1.20–1.55)** | 294 | 307.27 (236.94–384.96) | 0.96 (0.85–1.07) | 349 | 363.94 (311.01–422.05) | 0.96 (0.86–1.06) | 275 | 325.47 (271.85–384.42) | 0.84 (0.75–0.94) |
| Jun | 237 | 158.91 (90.94–233.16) | **1.49 (1.30–1.68)** | 271 | 277.42 (197.45–349.29) | 0.98 (0.86–1.09) | 314 | 327.66 (270.00–358.24) | 0.96 (0.85–1.06) | 313 | 309.47 (250.46–367.39) | 1.01 (0.90–1.12) |
| Jul | 253 | 191.27 (124.03–264.49) | **1.32 (1.16–1.49)** | 326 | 328.83 (255.82–403.36) | 0.99 (0.88–1.10) | 333 | 381.03 (320.80–439.07) | 0.87 (0.78–0.97) | 316 | 332.84 (275.08–390.19) | 0.95 (0.84–1.05) |
| Aug | 211 | 169.78 (103.41–235.10) | **1.24 (1.08–1.41)** | 295 | 326.84 (255.61–400.31) | 0.90 (0.80–1.01) | 332 | 370.65 (312.02–424.35) | 0.90 (0.80–0.99) | 347 | 305.97 (252.25–361.93) | **1.13 (1.01–1.25)** |
| Sep | 210 | 150.78 (83.74–218.61) | **1.39 (1.20–1.58)** | 254 | 277.47 (200.91–355.99) | 0.92 (0.80–1.03) | 272 | 343.59 (287.20–400.41) | 0.79 (0.70–0.89) | 284 | 282.35 (226.67–340.70) | 1.01 (0.89–1.12) |
| Oct | 218 | 148.00 (76.29–216.79) | **1.47 (1.28–1.67)** | 276 | 269.42 (196.29–350.41) | 1.02 (0.90–1.15) | 327 | 319.79 (264.02–377.21) | 1.02 (0.91–1.13) | 326 | 287.58 (231.51–342.28) | **1.13 (1.01–1.26)** |
| Nov | 238 | 211.55 (144.73–279.37) | 1.13 (0.98–1.27) | 255 | 325.91 (250.47–398.07) | 0.78 (0.69–0.88) | 285 | 330.73 (275.49–386.75) | 0.86 (0.76–0.96) | 283 | 261.78 (203.19–314.49) | 1.08 (0.96–1.21) |
| Dec | 168 | 170.12 (102.19–241.82) | 0.99 (0.84–1.14) | 270 | 263.24 (187.53–343.28) | 1.03 (0.90–1.15) | 258 | 299.53 (243.78–353.85) | 0.86 (0.76–0.97) | 217 | 221.55 (164.34–275.65) | 0.98 (0.85–1.11) |

Bold indicates a statistically significant OER with a 95% CI that excludes the null value of 1.00.

Abbreviations: OER, Observed-to-Expected Ratio; CI, Confidence Interval.

number of suicides per month during the pandemic was lower among those aged 50–64 years (Fig 3(C) and Table 3), but significantly higher among those aged ≤ 34 years (Fig 3(A) and Table 3). The monthly number of suicides by those aged 35–49 (Fig 3(B) and Table 3) and ≥ 65 (Fig 3(D) and Table 3) years was largely in line with the projections.

As supporting information, we present the trends calculated by Prophet models based on data from 1997 to 2018, as well as the monthly number of suicides to show suicide trends in Korea over the past two decades (S1–S3 Figs). According to the suicide trends through 2018, the number of suicides peaked in 2010–2012, and then declined gradually for both men and women (S1 and S2 Figs), and for all age groups except 35–49 years (S3 Fig).

### 3.2. Suicide deaths per year before and after the COVID-19 outbreak

We present the number of suicides and suicide rate per year between 2017 and 2020 (Table 4). The number of suicides and suicide rate per 100,000 population increased in 2018 and 2019 compared to 2017, but declined again in 2020. This pattern was also observed in male suicides and suicides by those aged ≥ 35 years. However, among women and those aged ≤ 34 years, suicides increased steadily from 2017, and the number and rate of suicides in 2020 also exceeded the projected estimates for that year.

## 4. Discussion

In this study, we determined how much the number of suicides during the pandemic differed from the expected number with the forecasting model. The forecasting models incorporated the effect of celebrity suicides on suicide trends and were validated as the expected number of suicides in 2019 closely matched the observed number in 2019. The total number of suicides per month during the COVID-19 pandemic did not deviate from the projections (Fig 1 and Table 1). However, there were differences in suicide trends among sex and age groups. During the pandemic, the monthly number of female suicides (Fig 2(B) and Table 2) and those aged ≤ 34 years (Fig 3(A) and Table 3) was significantly higher than expected.

The suicide rate in Korea fluctuates in the short and long term, although Korea has a high suicide rate. As suicide has emerged as a serious health problem in Korea, the Korean government implemented national suicide prevention policies in the early 2000s [20]. As a result, the number of suicide deaths in Korea has decreased gradually since 2011–12 [21]. Furthermore, the suicide trends are characterized by seasonal variation, with more suicides in spring and a significant decrease in winter [22]. They are also affected by copycat suicides, the phenomenon of committing suicide immediately following suicide by celebrities [23]. Considering the fluctuations in suicide trends, it might be inappropriate to compare the number of suicides during the COVID-19 pandemic directly with that of the previous years. Therefore, we first developed forecasting models based on data for the monthly number of suicides between 1997 and 2018 to predict the number of suicides in the pandemic. Then, we compared the expected and actual numbers of suicides to determine whether the COVID-19 pandemic changed suicide trends in Korea.

### 4.1. Suicide deaths during the COVID-19 pandemic

The COVID-19 pandemic did not affect the total number of suicides in Korea significantly, consistent with recent reports of no significant change in suicide trends after the COVID-19 outbreak in Western countries [9, 10]. In the early period of the COVID-19 pandemic, experts were concerned that suicide risk could have increased significantly due to declining incomes, social isolation, disrupted mental health delivery, and increased sales of alcohol and arms [8]. However, evidence shows that the COVID-19 pandemic has not had a significant impact on

the suicide trends, at least in the early period [11]. The prevalence of COVID-19 also does not seem to correlate with an increase in suicide rates. As a plausible explanation for this, we hypothesized that the worldwide public health crisis overwhelming individuals and societies may paradoxically suppress the incidence of suicides. Emotional problems caused by personal stress can increase suicidality. Suicide rates tend to be decreased by serious external threats, such as war and natural disasters [24]. From an evolutionary perspective, the instinct to protect oneself from external danger might decrease suicidality. However, after passing through the acute crisis, suicidality can increase, along with emotional distress [25]. Therefore, we should take care because suicide rates might increase in the near future with long-lasting economic and social troubles [26].

## 4.2. Male and female suicides during the COVID-19 pandemic

Our results show that the COVID-19 pandemic had a greater impact on female than male suicides. The monthly number of female suicides in 2020 was significantly higher than expected (Fig 2(B) and Table 2), but the number of male suicides in 2020 did not deviate from the projection (Fig 2(A) and Table 2). In addition, the annual number of female suicides in 2020 was similar to that in 2019, and the number of male suicides in 2020 was lower than in 2019 (Table 4). In Korea, female suicides account for about one-fourth of all suicides, and were declining steadily until 2017 [21]. However, the suicide rates among women started to increase following a series of celebrity suicides at the end of 2019, and the increasing trend continued during the COVID-19 pandemic (Fig 2(B)), halting the downward trend over the past decade (S2 Fig). These findings are in line with recent studies showing that female suicides in Japan increased significantly after the first wave of the COVID-19 outbreak, suggesting that the COVID-19 pandemic had a more devastating effect on suicides among women [27, 28]. Several studies have shown that women are more vulnerable to the impact of the COVID-19 pandemic on mental health [14, 29]. Previously, we also demonstrated that women felt more psychological distress due to restrictions on social activities during the COVID-19 pandemic, while men have suffered mainly from financial difficulties [30]. Since social interaction is more crucial for coping with stress in women, prolonged social isolation and loneliness may have played an important role in the increase in suicides among women. Conversely, large-scale economic support policies for the low-income class might have prevented male suicides to an extent. Although we do not fully understand the mechanism behind the upward trend in female suicides during the pandemic, the findings suggest that there are differences in suicide risk factors in the pandemic between men and women and further studies need to address the sex difference in suicides in the COVID-19 pandemic.

**Table 4. Suicides per year in Korea from 2017 to 2020.**

| Year | 2017 | 2018 | 2019 | 2020 |
|---|---|---|---|---|
| Total | 12,463 (24.3) | 13,670 (26.6) | 13,799 (26.9) [13,548.33 (26.4)] * | 13,195 (25.7) [13,000.94 (25.3)] |
| Men | 8,922 (34.9) | 9,862 (38.5) | 9,730 (38.0) [9,585.87 (37.4)] | 9,093 (35.5) [9,356.38 (36.5)] |
| Women | 3,541 (13.8) | 3,808 (14.8) | 4,069 (15.8) [4,059.31 (15.8)] | 4,102 (15.9) [3,709.20 (14.4)] |
| Age ≤ 34 | 2,150 (10.7) | 2,320 (11.8) | 2,380 (12.4) [2,304.41 (12.0)] | 2,597 (13.8) [2,060.88 (10.9)] |
| Age 35–49 | 3,431 (27.1) | 3,847 (30.7) | 3,728 (30.3) [3,714.21 (30.2)] | 3,470 (28.8) [3,538.26 (29.4)] |
| Age 50–64 | 3,508 (30.5) | 3,910 (33.1) | 4,085 (33.7) [4,015.71 (33.1)] | 3,736 (30.3) [4,064.02 (32.9)] |
| Age ≥ 65 | 3,372 (47.7) | 3,593 (48.6) | 3,600 (46.6) [3,481.45 (45.1)] | 3,392 (41.7) [3,395.28 (41.7)] |

Data are shown as the number of suicides (suicides per 100,000 population).

* indicates the sum of the expected number of suicides per month according to forecasting models (with 2019 as the validation period).

### 4.3. Suicides by age group during the COVID-19 pandemic

The number of suicide deaths during the COVID-19 pandemic showed different patterns by age group. Compared to the projection for 2020, the monthly number of suicides by those aged ≤ 34 years was significantly higher (Fig 3(A) and Table 3), while the number of suicides by those aged 50–64 years was lower (Fig 3(C) and Table 3). In addition, compared to 2019, the annual number of suicides by those aged ≤ 34 years increased in 2020, while the number of suicides decreased in all other age groups (Table 4). In Korea, the number of suicides by those aged ≤ 34 years has been lower than that of the other age groups, and has shown a declining trend since 2011–12 (S3A Fig). By contrast, the number of suicides was highest among those aged 50–64 years in recent years, and has steadily increased in the 2010s (S3C Fig). Against this background, the different trends in suicide during the COVID-19 pandemic among age groups suggest that pandemics promote suicide among young people but suppress it among middle-aged people. Recent studies have found that young people were more vulnerable to mental health problems, including depression, during the COVID-19 pandemic [31, 32]. In addition, there have been reports of an increase in suicide attempts and suicidal thoughts among adolescents and young adults during the pandemic [33, 34]. For young people who are less vulnerable to COVID-19 infection, the restrictions in daily and economic activities due to quarantine policies might have increased their psychological burden, reflected in loneliness, depression, and despair [35–37]. They may also experience greater fear and worry about the future while living in a socioeconomically unstable society [31]. The suicide trend among young people during the COVID-19 pandemic is unclear, and further investigation of their suicidal behaviors and its risk factors is needed.

### 4.4. Limitations

This study had some methodological limitations. First, we used only one algorithm, Prophet, as a forecasting procedure. Although the forecasting performance of the Prophet algorithm was validated using the data for 2019 in this study, additional forecasting models are needed to find a better procedure to predict suicide trends. Second, we could not address how significantly the suicide trends changed after the COVID-19 outbreak compared to past years. As a next step, we plan to conduct an interrupted time-series analysis to verify the changes in suicide trends caused by the pandemic in Korea. Third, considering potential suicide deaths that have not been officially confirmed, the impact of the COVID-19 pandemic on suicide in Korea may have been underestimated in this study. Therefore, further research on unintentional deaths and those of undetermined intent, as well as self-harm and suicidal behaviors during the pandemic, may provide a more accurate picture [38]. Fourth, this study did not address differences in suicide incidence by educational level, economic status, or region. It is necessary to investigate suicide trends during the pandemic using an integrated model analyzing the effect of multiple risk factors, including demographic characteristics and socioeconomic conditions.

### 4.5. Conclusion

In conclusion, the COVID-19 pandemic has not increased the total number of suicide deaths in Korea. However, the number of female and youth suicides during the pandemic increased. This suggests that the COVID-19 pandemic might have increased the suicide risk among women and young people in Korea. Further studies of the long-term trends in suicides and related socioeconomic factors during the COVID-19 pandemic are warranted.

## Supporting information

**S1 Table. Performance metrics for forecasting models.**
(DOCX)

**S1 Fig. Trends in the numbers of suicides per month between 1997 and 2018 in Korea.** The dots represent the numbers of suicides and the red dashed line indicates the trend. The blue vertical dotted lines indicate the dates of celebrity suicides.
(TIF)

**S2 Fig. Trends in the numbers of male and female suicides per month between 1997 and 2018 in Korea.** The dots represent the numbers of suicides and the red dashed line indicates the trend. The blue vertical dotted lines indicate the dates of celebrity suicides.
(TIF)

**S3 Fig. Trends in the numbers of suicides by age group per month between 1997 and 2018 in Korea.** The dots represent the numbers of suicides and the red dashed line indicates the trend. The blue vertical dotted lines indicate the dates of celebrity suicides.
(TIF)

## Author Contributions

**Conceptualization:** Seunghyong Ryu, Hee Jung Nam, Sung-Wan Kim.

**Data curation:** Seunghyong Ryu, Hee Jung Nam, Min Jhon.

**Formal analysis:** Seunghyong Ryu, Min Jhon, Sung-Wan Kim.

**Funding acquisition:** Sung-Wan Kim.

**Visualization:** Min Jhon.

**Writing – original draft:** Seunghyong Ryu, Hee Jung Nam, Sung-Wan Kim.

**Writing – review & editing:** Ju-Yeon Lee, Jae-Min Kim.

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
