## [Decision Letter · Decision Letter 0]

3 May 2022

PONE-D-22-03951Trends in suicide deaths before and after the COVID-19 outbreak in KoreaPLOS ONE

Dear Dr. Kim,

Thank you for submitting your manuscript to PLOS ONE. After careful consideration, we feel that it has merit but does not fully meet PLOS ONE’s publication criteria as it currently stands. Therefore, we invite you to submit a revised version of the manuscript that addresses the points raised during the review process.

We look forward to receiving your revised manuscript.

Kind regards,

Marco Innamorati

Academic Editor

PLOS ONE

Journal Requirements:

2. Please upload a new copy of Figure 3 and 6 as the detail is not clear. Please follow the link for more information: https://blogs.plos.org/plos/2019/06/looking-good-tips-for-creating-your-plos-figures-graphics/

Reviewers' comments:

Reviewer's Responses to Questions

**Comments to the Author**

1. Is the manuscript technically sound, and do the data support the conclusions?

Reviewer #1: Yes

Reviewer #2: Partly

2. Has the statistical analysis been performed appropriately and rigorously? 

Reviewer #1: I Don't Know

Reviewer #2: Yes

3. Have the authors made all data underlying the findings in their manuscript fully available?

Reviewer #1: Yes

Reviewer #2: Yes

4. Is the manuscript presented in an intelligible fashion and written in standard English?

Reviewer #1: Yes

Reviewer #2: Yes

5. Review Comments to the Author

Reviewer #1: The authors present sex- and age-specific suicide numbers observed in (South?) Korea between Jan 2019 and Dec 2020 and compared them with the expected values based on a forecasting procedure with data reaching back as far as 1997.

Currently we see a huge number of publications with national suicide data in relation with COVID-19. South Korea is one of the countries that deserve a somewhat higher interest given the neighborhood to Japan where suicide data seem to deviate from the world-wide development during the pandemic.

Thus, the study is generally worth to be considered for publication. However, I have two major comments.

• I think in April 2022 it is reasonable to wait for the data of 2021 and include them, too. I guess these should be available in the near future and recommend to concede the authors a sufficiently long time frame for resubmission.

• I don't see whether the Prophet tool takes the development of the general population number into account. If not so, I would strongly recommend to use suicide rates instead of numbers.

Minor points:

• Page 6, line 84 says that the blue line represents the expected numbers for 2020 but in the figures it starts already in Jan 2019.

• The legends in the figures are blurred and difficult to read.

• Figg. 4-6: why are the data of 2019 and 2020 not displayed?

• Moreover, the many dots do not contribute much to the information of importance and give a busy picture. Annually rates would in my opinion be better. Then, the necessary information could be presented as 7 lines in one figure and Figg. 5 and 6 and Table 4 could be saved.

Reviewer #2: The paper addresses the issue of putative Covid-19 pandemic mental health effects on suicide trends in Korea and concludes that there was no global increase of suicides though an increase is observable between women and young people below 34.

The introduction and review of literature is adequate and balanced and the study is highly relevant considering the high suicide rates in Korea.

The procedural rational seems theoretically correct: comparing observed suicide with forecast based on 20-years trend resorting to time series analysis, similar to ARIMA (Prophet allowing to adjust seasonality and holidays).

However, in the following papers, using joinpoint analysis, trend changes occurred in 1990 for both men and women and in 2004 for men and 2009 for women and men.

Park, C., Jee, Y. H., & Jung, K. J. (2016). Age–period–cohort analysis of the suicide rate in Korea. Journal of Affective Disorders, 194, 16-20. https://doi.org/10.1016/j.jad.2016.01.021

Lee S, Park J, Lee S, et al Changing trends in suicide rates in South Korea from 1993 to 2016: a descriptive study BMJ Open 2018;8:e023144. doi: 10.1136/bmjopen-2018-023144

Joinpoint tells us nothing on structural integrity of data and if the breakpoints are true trend changes due to preventive measures or registry errors. The present study uses trend analysis for the purpose of forecasting. Shouldn’t the quality of suicide data registry and trending between 1997 and 2019 be assessed in order to consider forecast effects?

A study using R Strucchange procedure on a national suicide long series could be useful for quality analysis:

Gusmão, R., Ramalheira, C., Conceição, V., Severo, M., Mesquita, E., Xavier, M., & Barros, H. (2021). Suicide time-series structural change analysis in Portugal (1913-2018): Impact of register bias on suicide trends. J Affect Disord, 291, 65-75. https://doi.org/10.1016/j.jad.2021.04.048

Also, on the discussion of this research we believe the ARIMA model should be included otherwise cherry-picking becomes suspicious.

6. PLOS authors have the option to publish the peer review history of their article (what does this mean?). If published, this will include your full peer review and any attached files.

Reviewer #1: No

Reviewer #2: No

---

## [Author Response · Author response to Decision Letter 0]

7 Jun 2022

Response to the Reviewers' comments:

Reviewer #1: The authors present sex- and age-specific suicide numbers observed in (South?) Korea between Jan 2019 and Dec 2020 and compared them with the expected values based on a forecasting procedure with data reaching back as far as 1997.

Currently we see a huge number of publications with national suicide data in relation with COVID-19. South Korea is one of the countries that deserve a somewhat higher interest given the neighborhood to Japan where suicide data seem to deviate from the world-wide development during the pandemic.

Thus, the study is generally worth to be considered for publication. However, I have two major comments.

- Response: We thank Reviewer #1 for the thoughtful suggestions.

I think in April 2022 it is reasonable to wait for the data of 2021 and include them, too. I guess these should be available in the near future and recommend to concede the authors a sufficiently long time frame for resubmission.

- Response: We examined the early impact of the COVID-19 pandemic on suicides in South Korea. The official number of suicides in 2020 was issued by Statistics Korea in October 2021. This study compared the official number of suicides in the early months of the pandemic with the expected number based on the pre-pandemic period.

We believe that data for a longer period would be needed to verify whether the suicide rate changed before and after the COVID-19 outbreak. According to Statistics Korea, the official number of suicides in 2021 should be released in the fall of 2022. As mentioned in our discussion, we plan to conduct a further study using an interrupted time-series design to detect breakpoints in suicide trends around the pandemic period.

I don't see whether the Prophet tool takes the development of the general population number into account. If not so, I would strongly recommend to use suicide rates instead of numbers.

- Response: We used Prophet models based on the monthly number of suicides without considering the total population size per month. As mentioned in the Introduction, the suicide rate in Korea has declined steadily since around 2010. Therefore, we thought that it would be inappropriate to compare the suicide rates before and after the COVID-19 outbreak directly. Instead, we compared the observed number of suicides per month in the early months of the pandemic with estimates from the Prophet model based on the pre-pandemic data. Comparing the monthly number of suicides over a year may be more sensitive than comparing monthly suicide rate per 100,000 population. Our method for comparing the observed and expected number of suicides largely followed that of Pirkis et al. (Suicide trends in the early months of the COVID-19 pandemic: an interrupted time-series analysis of preliminary data from 21 countries. Lancet Psychiatry 2021, 8(7), 579–588).

In addition, we have presented the annual number and rate of suicides in recent years in Section 3.2. and Table 4 (highlighted, page 10 line 115–121).

Minor points:

Page 6, line 84 says that the blue line represents the expected numbers for 2020 but in the figures it starts already in Jan 2019.

- Response: In Figs. 1–3, we denote the expected number of suicides for 2019 and 2020, according to forecasting models, with a blue line. First, we show the expected number for 2019 to demonstrate how closely it matches the observed number (i.e., model accuracy). The expected number for 2020 is then provided to show the extent of the deviation from the observed number of suicides for 2020. We have corrected the legend and stated that data for 2019 were used as the validation set in Figs. 1–3.

The legends in the figures are blurred and difficult to read.

- Response: We have made the lines and annotations clearer.

Fig. 4-6: why are the data of 2019 and 2020 not displayed? Moreover, the many dots do not contribute much to the information of importance and give a busy picture. Annually rates would in my opinion be better. Then, the necessary information could be presented as 7 lines in one figure and Fig. 5 and 6 and Table 4 could be saved.

- Response: Figures 4–6 show the monthly number and trend of suicides between 1997 and 2018; the forecasting models were trained on these data. In the original manuscript, we presented these data only to demonstrate that the number of suicides in Korea declined steadily from around 2010 until the COVID-19 outbreak. However, the declining trend of suicides in Korea, which has been reported in other studies, was mentioned in the Introduction of this manuscript and we do not consider these data to be the main results. Therefore, Figs. 4–6 in the original manuscript are Supplementary Figs. 1–3 in the revised manuscript (highlighted, page 6 line 86–90). Instead, following the reviewer’s recommendation, we provide the annual suicide rate (per 100,000 population) and number of suicides per year in recent years in Section 3.2 and Table 4 (highlighted, page 10 line 115–121).

Reviewer #2: The paper addresses the issue of putative Covid-19 pandemic mental health effects on suicide trends in Korea and concludes that there was no global increase of suicides though an increase is observable between women and young people below 34. 

The introduction and review of literature is adequate and balanced and the study is highly relevant considering the high suicide rates in Korea.

The procedural rational seems theoretically correct: comparing observed suicide with forecast based on 20-years trend resorting to time series analysis, similar to ARIMA (Prophet allowing to adjust seasonality and holidays).

- Response: We thank Reviewer #2 for the thoughtful suggestions.

However, in the following papers, using joinpoint analysis, trend changes occurred in 1990 for both men and women and in 2004 for men and 2009 for women and men.

Park, C., Jee, Y. H., & Jung, K. J. (2016). Age–period–cohort analysis of the suicide rate in Korea. Journal of Affective Disorders, 194, 16-20. https://doi.org/10.1016/j.jad.2016.01.021

Lee S, Park J, Lee S, et al Changing trends in suicide rates in South Korea from 1993 to 2016: a descriptive study BMJ Open 2018;8:e023144. doi: 10.1136/bmjopen-2018-023144

Joinpoint tells us nothing on structural integrity of data and if the breakpoints are true trend changes due to preventive measures or registry errors. The present study uses trend analysis for the purpose of forecasting. Shouldn’t the quality of suicide data registry and trending between 1997 and 2019 be assessed in order to consider forecast effects?

A study using R Strucchange procedure on a national suicide long series could be useful for quality analysis:

Gusmão, R., Ramalheira, C., Conceição, V., Severo, M., Mesquita, E., Xavier, M., & Barros, H. (2021). Suicide time-series structural change analysis in Portugal (1913-2018): Impact of register bias on suicide trends. J Affect Disord, 291, 65-75. https://doi.org/10.1016/j.jad.2021.04.048

- Response: We appreciate the reviewer’s comments and agree fully that it is necessary to check for potential registration bias in suicide statistics in South Korea. In addition, given the possibility of underestimating the number of suicides, as mentioned in the Discussion, we need to consider presumed suicides and deaths with an undetermined cause that have not been officially reported. However, investigating these factors would require further analyses of other external cause of death statistics, or the recommended Joinpoint analysis. We consider that it is beyond the scope of this study to investigate the extent to which the number of suicides in the early months of the pandemic deviated from the projections based on the pre-pandemic period.

The suicide trends shown in Figs. 4–6, which were calculated from forecasting models trained on data from 1997 to 2018, were not the main study findings. In the original manuscript, we presented Figs. 4–6 only to demonstrate that the number of suicides in Korea had declined steadily from around 2010 until the COVID-19 outbreak (highlighted, page 6 line 86–90).

To avoid confusion, Figs. 4–6 in the original manuscript are Supplementary Figs. 1–3 in the revised manuscript. In addition, according to the reviewer’s recommendation, we plan to conduct further studies of registration bias in suicide statistics, as well as of unintentional deaths and those of underdetermined intent in Korea (highlighted, page 14 line 223–224).

Also, on the discussion of this research we believe the ARIMA model should be included otherwise cherry-picking becomes suspicious.

- Response: As shown in Figs. 1–3, there were several suicides by celebrities in Korea before and after the COVID-19 outbreak. We suspected that these celebrity suicides might have had a significant impact on suicide trends during the early stage of the pandemic. Therefore, to adjust for the effect of celebrity suicides, we used the Prophet algorithm, which has the advantage of being able to adjust for the effect of holidays and other recurring events. We have explained why we used the Prophet algorithm in Section 2.2 (highlighted, page 5 line 53–55).

In addition, following the reviewer’s recommendation, we compared the accuracy in predicting suicides in 2019 (validation period) between the Prophet and ARIMA algorithms, and found that the fitted Prophet models achieved comparable, or better, performance than the ARIMA models (highlighted, page 5 line 62–64). The RMSE, MAE, and MAPE values of the Prophet and ARIMA models calculated from the observed and expected numbers of suicides in 2019 (validation period) are provided in Supplementary Table 1.

---

## [Decision Letter · Decision Letter 1]

4 Aug 2022

PONE-D-22-03951R1Trends in suicide deaths before and after the COVID-19 outbreak in KoreaPLOS ONE

Dear Dr. Kim,

Thank you for submitting your manuscript to PLOS ONE. After careful consideration, we feel that it has merit but does not fully meet PLOS ONE’s publication criteria as it currently stands. Therefore, we invite you to submit a revised version of the manuscript that addresses the points raised during the review process.

I support the opinion of the reviewer who requested to wait with publication until data of 2021 can be included. Alternatively, include in a cover letter why you think that the paper could be interesting for readers of PLOS ONE in it's present form.

We look forward to receiving your revised manuscript.

Kind regards,

Marco Innamorati

Academic Editor

PLOS ONE

Reviewers' comments:

Reviewer's Responses to Questions

**Comments to the Author**

1. If the authors have adequately addressed your comments raised in a previous round of review and you feel that this manuscript is now acceptable for publication, you may indicate that here to bypass the “Comments to the Author” section, enter your conflict of interest statement in the “Confidential to Editor” section, and submit your "Accept" recommendation.

Reviewer #1: (No Response)

Reviewer #2: All comments have been addressed

Reviewer #3: All comments have been addressed

2. Is the manuscript technically sound, and do the data support the conclusions?

Reviewer #1: Partly

Reviewer #2: Yes

Reviewer #3: Yes

3. Has the statistical analysis been performed appropriately and rigorously? 

Reviewer #1: Yes

Reviewer #2: Yes

Reviewer #3: Yes

4. Have the authors made all data underlying the findings in their manuscript fully available?

Reviewer #1: Yes

Reviewer #2: Yes

Reviewer #3: Yes

5. Is the manuscript presented in an intelligible fashion and written in standard English?

Reviewer #1: Yes

Reviewer #2: Yes

Reviewer #3: Yes

6. Review Comments to the Author

Reviewer #1: (No Response)

Reviewer #2: Non applicable right now: the authors answers to the first review meet all my doubts and therefore I suggest immediate publication.

Reviewer #3: This is, in summary, an interesting study aimed to investigate the effect of the coronavirus disease-2019 (COVID-19) pandemic on suicide trends in Korea via a time-series analysis. The authors found that the total numbers of suicides during the COVID-19 pandemic did not deviate from projections based on the pre-pandemic period. In addition, the number of suicides among females and those under the age of 34 years significantly exceeded the expected level. Moreover, the COVID-19 pandemic did not increase the overall suicide rate significantly. Finally, suicides among women and young people increased, suggesting that the pandemic might drive more members of these groups to suicide

Overall, the present manuscript is interesting and well-written in its current version; thus, only minor changes are required, in my opinion.

The authors may find my minor comments below.

First, when throughout the Introduction section, the authors correctly stressed the psychosocial impairment and disability in the context of COVID-19, they might further stress the impact of Covid-19 related lockdown on lifestyle habits and behavioral risk factors. Importantly, the impact of COVID-19 lockdown on physical, mental, and social wellbeing of elderly and fragile populations in specific countries such as Italy cannot be ignored. The multi-disciplinary competencies together with appropriate funding and access to rich data sources may allow to fulfill interesting research objectives. Thus, according to this background, the study of Odone and coworkers published on Acta Biomed (PMID: 32701921) may be cited within the main text.

In addition, the authors might further mention, in the context of Covid-19 infection, the link between the deterioration of people’s mental health and the consequent increase in suicide rate which is frequently underreported. Unfortunately, above 2% of the traffic accidents are suicide behaviors. This phenomenon may be underreported considering that suicides by car accidents may be reported as accidental in the national statistics. Therefore, given the above information, my suggestion is to include within the manuscript, the study published in 2016 on Forensic Sci Int (PMID: 22576104).

7. PLOS authors have the option to publish the peer review history of their article (what does this mean?). If published, this will include your full peer review and any attached files.

Reviewer #1: No

Reviewer #2: No

Reviewer #3: No

---

## [Author Response · Author response to Decision Letter 1]

8 Aug 2022

Response to the Reviewers' comments:

Reviewer #3: This is, in summary, an interesting study aimed to investigate the effect of the coronavirus disease-2019 (COVID-19) pandemic on suicide trends in Korea via a time-series analysis. The authors found that the total numbers of suicides during the COVID-19 pandemic did not deviate from projections based on the pre-pandemic period. In addition, the number of suicides among females and those under the age of 34 years significantly exceeded the expected level. Moreover, the COVID-19 pandemic did not increase the overall suicide rate significantly. Finally, suicides among women and young people increased, suggesting that the pandemic might drive more members of these groups to suicide

Overall, the present manuscript is interesting and well-written in its current version; thus, only minor changes are required, in my opinion.

- Response: We thank Reviewer #3 for the thoughtful suggestions.

First, when throughout the Introduction section, the authors correctly stressed the psychosocial impairment and disability in the context of COVID-19, they might further stress the impact of Covid-19 related lockdown on lifestyle habits and behavioral risk factors. Importantly, the impact of COVID-19 lockdown on physical, mental, and social wellbeing of elderly and fragile populations in specific countries such as Italy cannot be ignored. The multi-disciplinary competencies together with appropriate funding and access to rich data sources may allow to fulfill interesting research objectives. Thus, according to this background, the study of Odone and coworkers published on Acta Biomed (PMID: 32701921) may be cited within the main text.

- Response: Following the reviewer’s comments, we have cited the Odone et al.’s study (PMID: 32701921) in the Introduction section (highlighted, page 3 line 9 – 10).

In addition, the authors might further mention, in the context of Covid-19 infection, the link between the deterioration of people’s mental health and the consequent increase in suicide rate which is frequently underreported. Unfortunately, above 2% of the traffic accidents are suicide behaviors. This phenomenon may be underreported considering that suicides by car accidents may be reported as accidental in the national statistics. Therefore, given the above information, my suggestion is to include within the manuscript, the study published in 2016 on Forensic Sci Int (PMID: 22576104).

- Response: Thanks for the reviewer’s comments. We have added the reference (PMID: 22576104) in the Limitation section (highlighted, page 14 line 223 – 224).

---

## [Decision Letter · Decision Letter 2]

12 Aug 2022

Trends in suicide deaths before and after the COVID-19 outbreak in Korea

PONE-D-22-03951R2

Dear Dr. Kim,

We’re pleased to inform you that your manuscript has been judged scientifically suitable for publication and will be formally accepted for publication once it meets all outstanding technical requirements.

Kind regards,

Marco Innamorati

Academic Editor

PLOS ONE

Additional Editor Comments (optional):

Reviewers' comments:

Reviewer's Responses to Questions

**Comments to the Author**

1. If the authors have adequately addressed your comments raised in a previous round of review and you feel that this manuscript is now acceptable for publication, you may indicate that here to bypass the “Comments to the Author” section, enter your conflict of interest statement in the “Confidential to Editor” section, and submit your "Accept" recommendation.

Reviewer #1: (No Response)

Reviewer #3: All comments have been addressed

2. Is the manuscript technically sound, and do the data support the conclusions?

Reviewer #1: (No Response)

Reviewer #3: Yes

3. Has the statistical analysis been performed appropriately and rigorously? 

Reviewer #1: (No Response)

Reviewer #3: Yes

4. Have the authors made all data underlying the findings in their manuscript fully available?

Reviewer #1: (No Response)

Reviewer #3: Yes

5. Is the manuscript presented in an intelligible fashion and written in standard English?

Reviewer #1: (No Response)

Reviewer #3: Yes

6. Review Comments to the Author

Reviewer #1: (No Response)

Reviewer #3: In the revised paper, the authors addressed most of the major questions raised by Reviewers improving both the main structure and quality of the present manuscript. I have no further additional comments.

7. PLOS authors have the option to publish the peer review history of their article (what does this mean?). If published, this will include your full peer review and any attached files.

Reviewer #1: No

Reviewer #3: No

---

## [Editor Report · Acceptance letter]

2 Sep 2022

PONE-D-22-03951R2 

Trends in suicide deaths before and after the COVID-19 outbreak in Korea 

Dear Dr. Kim:

I'm pleased to inform you that your manuscript has been deemed suitable for publication in PLOS ONE. Congratulations! Your manuscript is now with our production department. 

Kind regards, 

on behalf of

Dr. Marco Innamorati 

Academic Editor

PLOS ONE